# Quadrature-based features for kernel approximation

**Marina Munkhoeva**[†]     **Yermek Kapushev**[†]     **Evgeny Burnaev**[†]     **Ivan Oseledets**[†,‡]

[†]Skolkovo Institute of Science and Technology
Moscow, Russia
[‡]Institute of Numerical Mathematics of the Russian Academy of Sciences
Moscow, Russia

## Abstract

We consider the problem of improving kernel approximation via randomized feature maps. These maps arise as Monte Carlo approximation to integral representations of kernel functions and scale up kernel methods for larger datasets. Based on an efficient numerical integration technique, we propose a unifying approach that reinterprets the previous random features methods and extends to better estimates of the kernel approximation. We derive the convergence behaviour and conduct an extensive empirical study that supports our hypothesis[1].

## 1   Introduction

Kernel methods proved to be an efficient technique in numerous real-world problems. The core idea of kernel methods is the kernel trick – compute an inner product in a high-dimensional (or even infinite-dimensional) feature space by means of a kernel function $k$:

$$k(\mathbf{x}, \mathbf{y}) = \langle \psi(\mathbf{x}), \psi(\mathbf{y}) \rangle, \tag{1}$$

where $\psi : \mathcal{X} \to \mathcal{F}$ is a non-linear feature map transporting elements of input space $\mathcal{X}$ into a feature space $\mathcal{F}$. It is a common knowledge that kernel methods incur space and time complexity infeasible to be used with large-scale datasets directly. For example, kernel regression has $\mathcal{O}(N^3 + Nd^2)$ training time, $\mathcal{O}(N^2)$ memory, $\mathcal{O}(Nd)$ prediction time complexity for $N$ data points in original $d$-dimensional space $\mathcal{X}$.

One of the most successful techniques to handle this problem, known as Random Fourier Features (RFF) proposed by [29], introduces a low-dimensional randomized approximation to feature maps:

$$k(\mathbf{x}, \mathbf{y}) \approx \hat{\boldsymbol{\Psi}}(\mathbf{x})^\top \hat{\boldsymbol{\Psi}}(\mathbf{y}). \tag{2}$$

This is essentially carried out by using Monte-Carlo sampling to approximate scalar product in (1). A randomized $D$-dimensional mapping $\hat{\boldsymbol{\Psi}}(\cdot)$ applied to the original data input allows employing standard linear methods, i.e. reverting the kernel trick. In doing so one reduces the complexity to that of linear methods, e.g. $D$-dimensional approximation admits $\mathcal{O}(ND^2)$ training time, $\mathcal{O}(ND)$ memory and $\mathcal{O}(N)$ prediction time.

It is well known that as $D \to \infty$, the inner product in (2) converges to the exact kernel $k(\mathbf{x}, \mathbf{y})$. Recent research [35; 14; 9] aims to improve the convergence of approximation so that a smaller $D$ can be used to obtain the same quality of approximation.

This paper considers kernels that allow the following integral representation

$$k(\mathbf{x}, \mathbf{y}) = \mathbb{E}_{p(\mathbf{w})} f_{\mathbf{xy}}(\mathbf{w}) = I(f_{\mathbf{xy}}), \quad p(\mathbf{w}) = \frac{1}{(2\pi)^{d/2}} e^{-\frac{\|\mathbf{w}\|^2}{2}}, \quad f_{\mathbf{xy}} = \phi(\mathbf{w}^\top \mathbf{x})\phi(\mathbf{w}^\top \mathbf{y}). \tag{3}$$

For example, the popular Gaussian kernel admits such representation with $f_{\mathbf{xy}}(\mathbf{w}) = \phi(\mathbf{w}^\top \mathbf{x})^\top \phi(\mathbf{w}^\top \mathbf{y})$, where $\phi(\cdot) = [\cos(\cdot) \quad \sin(\cdot)]^\top$.

The class of kernels admitting the form in (3) covers shift-invariant kernels (e.g. radial basis function (RBF) kernels) and Pointwise Nonlinear Gaussian (PNG) kernels. They are widely used in practice and have interesting connections with neural networks [8; 34].

The main challenge for the construction of low-dimensional feature maps is the approximation of the expectation in (3) which is $d$-dimensional integral with Gaussian weight. Unlike other research studies we refrain from using simple Monte Carlo estimate of the integral, instead, we propose to use specific quadrature rules. We now list our contributions:

- We propose to use spherical-radial quadrature rules to improve kernel approximation accuracy. We show that these quadrature rules generalize the RFF-based techniques. We also provide an analytical estimate of the error for the used quadrature rules that implies better approximation quality.

- We use structured orthogonal matrices (so-called *butterfly matrices*) when designing quadrature rule that allow fast matrix by vector multiplications. As a result, we speed up the approximation of the kernel function and reduce memory requirements.

- We carry out an extensive empirical study comparing our methods with the state-of-the-art ones on a set of different kernels in terms of both kernel approximation error and downstream tasks performance. The study supports our hypothesis on the exceeding accuracy of the method.

## 2   Quadrature Rules and Random Features

We start with rewriting the expectation in Equation (3) as integral of $f_{\mathbf{xy}}$ with respect to $p(\mathbf{w})$:

$$I(f_{\mathbf{xy}}) = (2\pi)^{-\frac{d}{2}} \int_{-\infty}^{\infty} \cdots \int_{-\infty}^{\infty} e^{-\frac{\mathbf{w}^\top \mathbf{w}}{2}} f_{\mathbf{xy}}(\mathbf{w}) d\mathbf{w}.$$

Integration can be performed by means of quadrature rules. The rules usually take a form of interpolating function that is easy to integrate. Given such a rule, one may sample points from the domain of integration and calculate the value of the rule at these points. Then, the sample average of the rule values would yield the approximation of the integral.

The connection between integral approximation and mapping $\psi$ is straightforward. In what follows we show a brief derivation of the quadrature rules that allow for an explicit mapping of the form: $\psi(\mathbf{x}) = [\ a_0\phi(0)\ a_1\phi(\mathbf{w}_1^\top \mathbf{x})\ \ldots\ a_D\phi(\mathbf{w}_D^\top \mathbf{x})\ ]$, where the choice of the weights $a_i$ and the points $\mathbf{w}_i$ is dictated by the quadrature.

We use the average of sampled quadrature rules developed by [18] to yield unbiased estimates of $I(f_{\mathbf{xy}})$. A change of coordinates is the first step to facilitate stochastic spherical-radial rules. Now, let $\mathbf{w} = r\mathbf{z}$, with $\mathbf{z}^\top \mathbf{z} = 1$, so that $\mathbf{w}^\top \mathbf{w} = r^2$ for $r \in [0, \infty]$, leaving us with (to ease the notation we substitute $f_{\mathbf{xy}}$ with $f$)

$$I(f) = (2\pi)^{-\frac{d}{2}} \int_{U_d} \int_0^{\infty} e^{-\frac{r^2}{2}} r^{d-1} f(r\mathbf{z}) dr d\mathbf{z} = \frac{(2\pi)^{-\frac{d}{2}}}{2} \int_{U_d} \int_{-\infty}^{\infty} e^{-\frac{r^2}{2}} |r|^{d-1} f(r\mathbf{z}) dr d\mathbf{z}, \quad (4)$$

$I(f)$ is now a double integral over the unit $d$-sphere $U_d = \{\mathbf{z} : \mathbf{z}^\top \mathbf{z} = 1, \mathbf{z} \in \mathbb{R}^d\}$ and over the radius. To account for both integration regions we apply a combination of spherical ($S$) and radial ($R$) rules known as spherical-radial ($SR$) rules. To provide an intuition how the rules work, here we briefly state and discuss their form[2].

**Stochastic radial rules** of degree $2l + 1$ $R(h) = \sum_{i=0}^{l} \hat{w}_i \frac{h(\rho_i) + h(-\rho_i)}{2}$ have the form of the weighted symmetric sums and approximate the infinite range integral $T(h) = \int_{-\infty}^{\infty} e^{-\frac{r^2}{2}} |r|^{d-1} h(r) dr$. Note

that when $h$ is set to the function $f$ of interest, $T(f)$ corresponds to the inner integral in (4). To get an unbiased estimate for $T(h)$, points $\rho_i$ are sampled from specific distributions. The weights $\hat{w}_i$ are derived so that the rule is exact for polynomials of degree $2l + 1$ and give unbiased estimate for other functions.

**Stochastic spherical rules** $S_{\mathbf{Q}}(s) = \sum_{j=1}^{p} \widetilde{w}_j s(\mathbf{Q}\mathbf{z}_j)$, where $\mathbf{Q}$ is a random orthogonal matrix, approximate an integral of a function $s(\mathbf{z})$ over the surface of unit $d$-sphere $U_d$, where $\mathbf{z}_j$ are points on $U_d$, i.e. $\mathbf{z}_j^\top \mathbf{z}_j = 1$. Remember that the outer integral in (4) has $U_d$ as its integration region. The weights $\widetilde{w}_j$ are stochastic with distribution such that the rule is exact for polynomials of degree $p$ and gives unbiased estimate for other functions.

**Stochastic spherical-radial rules** $SR$ of degree $(2l + 1, p)$ are given by the following expression

$$SR_{\mathbf{Q},\rho}^{(2l+2,p)} = \sum_{j=1}^{p} \widetilde{w}_j \sum_{i=1}^{l} \hat{w}_i \frac{f(\rho\mathbf{Q}\mathbf{z}_i) + f(-\rho\mathbf{Q}\mathbf{z}_i)}{2},$$

where the distributions of weights are such that if degrees of radial rules and spherical rules coincide, i.e. $2l + 1 = p$, then the rule is exact for polynomials of degree $2l + 1$ and gives unbiased estimate of the integral for other functions.

## 2.1 Spherical-radial rules of degree $(1, 1)$ is RFF

If we take radial rule of degree 1 and spherical rule of degree 1, we obtain the following rule $SR_{\mathbf{Q},\rho}^{(1,1)} = \frac{f(\rho\mathbf{Q}\mathbf{z}) + f(-\rho\mathbf{Q}\mathbf{z})}{2}$, where $\rho \sim \chi(d)$. It is easy to see that $\rho\mathbf{Q}\mathbf{z} \sim \mathcal{N}(0, \mathbf{I})$, and for shift invariant kernel $f(\mathbf{w}) = f(-\mathbf{w})$, thus, the rule reduces to $SR_{\mathbf{Q},\rho}^{(1,1)} = f(\mathbf{w})$, where $\mathbf{w} \sim \mathcal{N}(0, \mathbf{I})$.

Now, RFF [29] makes approximation of the RBF kernel in exactly the same way: it generates random vector from Gaussian distribution and calculates the corresponding feature map.

**Proposition 2.1.** *Random Fourier Features for RBF kernel are SR rules of degree $(1, 1)$.*

## 2.2 Spherical-radial rules of degree $(1, 3)$ is ORF

Now, let's take radial rule of degree 1 and spherical rule of degree 3. In this case we get the following spherical-radial rule $SR_{\mathbf{Q},\rho}^{1,3} = \sum_{i=1}^{d} \frac{f(\rho\mathbf{Q}\mathbf{e}_i) + f(-\rho\mathbf{Q}\mathbf{e}_i)}{2}$, where $\rho \sim \chi(d)$, $\mathbf{e}_i = (0, \ldots, 0, 1, 0, \ldots, 0)^\top$ is an $i$-th column of the identity matrix.

Let us compare $SR^{1,3}$ rules with Orthogonal Random Features [14] for the RBF kernel. In the ORF approach, the weight matrix $\mathbf{W} = \mathbf{S}\mathbf{Q}$ is generated, where $\mathbf{S}$ is a diagonal matrix with the entries drawn independently from $\chi(d)$ distribution and $\mathbf{Q}$ is a random orthogonal matrix. The approximation of the kernel is then given by $k_{\text{ORF}}(\mathbf{x}, \mathbf{y}) = \sum_{i=1}^{d} f(\mathbf{w}_i)$, where $\mathbf{w}_i$ is the $i$-th row of the matrix $\mathbf{W}$. As the rows of $\mathbf{Q}$ are orthonormal, they can be represented as $\mathbf{Q}\mathbf{e}_i$.

**Proposition 2.2.** *Orthogonal Random Features for RBF kernel are SR rules of degree $(1, 3)$.*

## 2.3 Spherical-radial rules of degree $(3, 3)$

We go further and take both spherical and radial rules of degree 3, where we use original and reflected vertices $\mathbf{v}_j$ of randomly rotated unit vertex regular $d$-simplex $\mathbf{V}$ as the points on the unit sphere

$$SR_{\mathbf{Q},\rho}^{3,3}(f) = \left(1 - \frac{d}{\rho^2}\right) f(\mathbf{0}) + \frac{d}{d+1} \sum_{j=1}^{d+1} \left[\frac{f(-\rho\mathbf{Q}\mathbf{v}_j) + f(\rho\mathbf{Q}\mathbf{v}_j)}{2\rho^2}\right], \tag{5}$$

where $\rho \sim \chi(d + 2)$. We apply (5) to the approximation of (4) by averaging the samples of $SR_{\mathbf{Q},\rho}^{3,3}$:

$$I(f) = \mathbb{E}_{\mathbf{Q},\rho}[SR_{\mathbf{Q},\rho}^{3,3}(f)] \approx \hat{I}(f) = \frac{1}{n} \sum_{i=1}^{n} SR_{\mathbf{Q}_i,\rho_i}^{3,3}(f), \tag{6}$$

where $n$ is the number of sampled $SR$ rules. Speaking in terms of the approximate feature maps, the new feature dimension $D$ in case of the quadrature based approximation equals $2n(d+1)+1$ as we sample $n$ rules and evaluate each of them at $2(d+1)$ random points and 1 zero point.

In this work we propose to modify the quadrature rule by generating $\rho_j \sim \chi(d+2)$ for each $\mathbf{v}_j$, i.e. $SR_{\mathbf{Q},\rho}^{3,3}(f) = \left(1 - \sum_{j=1}^{d+1} \frac{d}{(d+1)\rho_j^2}\right) f(\mathbf{0}) + \frac{d}{d+1} \sum_{j=1}^{d+1} \left[\frac{f(-\rho_j \mathbf{Q}\mathbf{v}_j)+f(\rho_j \mathbf{Q}\mathbf{v}_j)}{2\rho_j^2}\right]$. It doesn't affect the quality of approximation while simplifies an analysis of the quadrature-based random features.

**Explicit mapping** We finally arrive at the map $\psi(\mathbf{x}) = [\ a_0\phi(0)\ \ a_1\phi(\mathbf{w}_1^\top \mathbf{x})\ \ \ldots\ \ a_D\phi(\mathbf{w}_D^\top \mathbf{x})\ ]$, where $a_0 = \sqrt{1 - \sum_{d+1}^{j=1} \frac{d}{\rho^2}}$ [3], $a_j = \frac{1}{\rho_j}\sqrt{\frac{d}{2(d+1)}}$, $\mathbf{w}_j$ is the $j$-th row in the matrix $\mathbf{W} = \boldsymbol{\rho} \otimes \left[\begin{smallmatrix} (\mathbf{Q}\mathbf{V})^\top \\ -(\mathbf{Q}\mathbf{V})^\top \end{smallmatrix}\right]$, $\boldsymbol{\rho} = [\rho_1 \ldots \rho_D]^\top$. To get $D$ features one simply stacks $n = \frac{D}{2(d+1)+1}$ such matrices $\mathbf{W}^k = \boldsymbol{\rho}^k \left[\begin{smallmatrix} (\mathbf{Q}^k\mathbf{V})^\top \\ -(\mathbf{Q}^k\mathbf{V})^\top \end{smallmatrix}\right]$ so that $\mathbf{W} \in \mathbb{R}^{D\times d}$, where only $\mathbf{Q}^k \in \mathbb{R}^{d\times d}$ and $\boldsymbol{\rho}^k$ are generated randomly ($k = 1, \ldots, n$). For Gaussian kernel, $\phi(\cdot) = [\cos(\cdot)\ \ \sin(\cdot)]^\top$. For the 0-order arc-cosine kernel, $\phi(\cdot) = \Theta(\cdot)$, where $\Theta(\cdot)$ is the Heaviside function. For the 1-order arc-cosine kernel, $\phi(\cdot) = \max(0, \cdot)$.

## 2.4 Generating uniformly random orthogonal matrices

The SR rules require a random orthogonal matrix $\mathbf{Q}$. If $\mathbf{Q}$ follows Haar distribution, the averaged samples of $SR_{\mathbf{Q},\rho}^{3,3}$ rules provide an unbiased estimate for (4). Essentially, Haar distribution means that all orthogonal matrices in the group are equiprobable, i.e. uniformly random. Methods for sampling such matrices vary in their complexity of generation and multiplication.

We test two algorithms for obtaining $\mathbf{Q}$. The first uses a QR decomposition of a random matrix to obtain a product of a sequence of reflectors/rotators $\mathbf{Q} = \mathbf{H}_1 \ldots \mathbf{H}_{n-1}\mathbf{D}$, where $\mathbf{H}_i$ is a random Householder/Givens matrix and a diagonal matrix $\mathbf{D}$ has entries such that $\mathbb{P}(d_{ii} = \pm 1) = \nicefrac{1}{2}$. It implicates no fast matrix multiplication. We test both methods for random orthogonal matrix generation and, since their performance coincides, we leave this one out for cleaner figures in the Experiments section.

The other choice for $\mathbf{Q}$ are so-called butterfly matrices [17]. For $d = 4$

$$\mathbf{B}^{(4)} = \begin{bmatrix} c_1 & -s_1 & 0 & 0 \\ s_1 & c_1 & 0 & 0 \\ 0 & 0 & c_3 & -s_3 \\ 0 & 0 & s_3 & c_3 \end{bmatrix} \begin{bmatrix} c_2 & 0 & -s_2 & 0 \\ 0 & c_2 & 0 & -s_2 \\ s_2 & 0 & c_2 & 0 \\ 0 & s_2 & 0 & c_2 \end{bmatrix} = \begin{bmatrix} c_1c_2 & -s_1c_2 & -c_1s_2 & s_1s_2 \\ s_1c_2 & c_1c_2 & -s_1s_2 & -c_1s_2 \\ c_3s_2 & -s_3s_2 & c_3c_2 & -s_3c_2 \\ s_3s_2 & c_3s_2 & s_3c_2 & c_3c_2 \end{bmatrix},$$

where $s_i,\ c_i$ is sine and cosine of some angle $\theta_i,\ i = 1, \ldots, d-1$. For definition and discussion please see Supplementary Materials. The factors of $\mathbf{B}^{(d)}$ are structured and allow fast matrix multiplication. The method using butterfly matrices is denoted by $\mathbf{B}$ in the Experiments section.

## 3 Error bounds

**Proposition 3.1.** *Let $l$ be a diameter of the compact set $\mathcal{X}$ and $p(\mathbf{w}) = \mathcal{N}(0, \sigma_p^2\mathbf{I})$ be the probability density corresponding to the kernel. Let us suppose that $|\phi(\mathbf{w}^\top \mathbf{x})| \leq \kappa$, $|\phi'(\mathbf{w}^\top \mathbf{x})| \leq \mu$ for all $\mathbf{w} \in \Omega$, $\mathbf{x} \in \mathcal{X}$ and $\left|\frac{1-f_{\mathbf{x}\mathbf{y}}(\rho\mathbf{z})}{\rho^2}\right| \leq M$ for all $\rho \in [0, \infty)$, where $\mathbf{z}^\top \mathbf{z} = 1$. Then for Quadrature-based Features approximation $\hat{k}(\mathbf{x}, \mathbf{y})$ of the kernel function $k(\mathbf{x}, \mathbf{y})$ and any $\varepsilon > 0$ it holds*

$$\mathbb{P}\left(\sup_{\mathbf{x},\mathbf{y}\in\mathcal{X}} |\hat{k}(\mathbf{x},\mathbf{y}) - k(\mathbf{x},\mathbf{y})| \geq \varepsilon\right) \leq \beta_d \left(\frac{\sigma_p l \kappa \mu}{\varepsilon}\right)^{\frac{2d}{d+1}} \exp\left(-\frac{D\varepsilon^2}{8M^2(d+1)}\right),$$

| Table 1: Space and time complexity. | | |
|---|---|---|
| **Method** | **Space** | **Time** |
| ORF | $\mathcal{O}(Dd)$ | $\mathcal{O}(Dd)$ |
| QMC | $\mathcal{O}(Dd)$ | $\mathcal{O}(Dd)$ |
| ROM | $\mathcal{O}(d)$ | $\mathcal{O}(d\log d)$ |
| **Quadrature based** | $\mathcal{O}(d)$ | $\mathcal{O}(d\log d)$ |

| Table 2: Experimental settings for the datasets. | | | | |
|---|---|---|---|---|
| **Dataset** | $N$ | $d$ | **#samples** | **#runs** |
| Powerplant | 9568 | 4 | 550 | 500 |
| LETTER | 20000 | 16 | 550 | 500 |
| USPS | 9298 | 256 | 550 | 500 |
| MNIST | 70000 | 784 | 550 | 100 |
| CIFAR100 | 60000 | 3072 | 50 | 50 |
| LEUKEMIA | 72 | 7129 | 10 | 10 |

*where $\beta_d = \left( d^{\frac{-d}{d+1}} + d^{\frac{1}{d+1}} \right) 2^{\frac{6d+1}{d+1}} \left( \frac{d}{d+1} \right)^{\frac{d}{d+1}}$. Thus we can construct approximation with error no more than $\varepsilon$ with probability at least $1 - \delta$ as long as*

$$D \geq \frac{8M^2(d+1)}{\varepsilon^2} \left[ \frac{2}{1 + \frac{1}{d}} \log \frac{\sigma_p l \kappa \mu}{\varepsilon} + \log \frac{\beta_d}{\delta} \right].$$

The proof of this proposition closely follows [33], details can be found in the Supplementary Materials.

Term $\beta_d$ depends on dimension $d$, its maximum is $\beta_{86} \approx 64.7 < 65$, and $\lim_{d\to\infty} \beta_d = 64$, though it is lower for small $d$. Let us compare this probability bound with the similar result for RFF in [33]. Under the same conditions the required number of samples to achieve error no more than $\varepsilon$ with probability at least $1 - \delta$ for RFF is the following

$$D \geq \frac{8(d+1)}{\varepsilon^2} \left[ \frac{2}{1 + \frac{1}{d}} \log \frac{\sigma_p l}{\varepsilon} + \log \frac{\beta_d}{\delta} + \frac{d}{d+1} \log \frac{3d+3}{2d} \right].$$

For Quadrature-based Features for RBF kernel $M = \frac{1}{2}, \kappa = \mu = 1$, therefore, we obtain

$$D \geq \frac{2(d+1)}{\varepsilon^2} \left[ \frac{2}{1 + \frac{1}{d}} \log \frac{\sigma_p l}{\varepsilon} + \log \frac{\beta_d}{\delta} \right].$$

The asymptotics is the same, however, the constants are smaller for our approach. See Section 4 for empirical justification of the obtained result.

**Proposition 3.2** ([33]). *Given a training set $\{(\mathbf{x}_i, y_i)\}_{i=1}^{n}$, with $\mathbf{x}_i \in \mathbb{R}^d$ and $y_i \in \mathbb{R}$, let $h(\mathbf{x})$ denote the result of kernel ridge regression using the positive semi-definite training kernel matrix $\mathbf{K}$, test kernel values $\mathbf{k_x}$ and regularization parameter $\lambda$. Let $\hat{h}(\mathbf{x})$ be the same using a PSD approximation to the training kernel matrix $\widehat{\mathbf{K}}$ and test kernel values $\hat{\mathbf{k}}_\mathbf{x}$. Further, assume that the training labels are centered, $\sum_{i=1}^{n} y_i = 0$, and let $\sigma_y^2 = \frac{1}{n} \sum_{i=1}^{n} y_i^2$. Also suppose $\|\mathbf{k_x}\|_\infty \leq \kappa$. Then*

$$|\hat{h}(\mathbf{x}) - h(\mathbf{x})| \leq \frac{\sigma_y \sqrt{n}}{\lambda} \|\hat{\mathbf{k}}_\mathbf{x} - \mathbf{k_x}\|_2 + \frac{\kappa \sigma_y n}{\lambda^2} \|\widehat{\mathbf{K}} - \mathbf{K}\|_2.$$

Suppose that $\sup |k(\mathbf{x}, \mathbf{x}') - \hat{k}(\mathbf{x}, \mathbf{x}')| \leq \varepsilon$ for all $\mathbf{x}, \mathbf{x}' \in \mathbb{R}^d$. Then $\|\hat{\mathbf{k}}_\mathbf{x} - \mathbf{k_x}\|_2 \leq \sqrt{n}\varepsilon$ and $\|\widehat{\mathbf{K}} - \mathbf{K}\|_2 \leq \|\widehat{\mathbf{K}} - \mathbf{K}\|_F \leq n\varepsilon$. By denoting $\lambda = n\lambda_0$ we obtain $|\hat{h}(\mathbf{x}) - h(\mathbf{x})| \leq \frac{\lambda_0+1}{\lambda_0^2} \sigma_y \varepsilon$. Therefore,

$$\mathbb{P} \left( |\hat{h}(\mathbf{x}) - h(\mathbf{x})| \geq \varepsilon \right) \leq \mathbb{P} \left( \|\hat{k}(\mathbf{x}, \mathbf{x}') - k(\mathbf{x}, \mathbf{x}')\|_\infty \geq \frac{\lambda_0^2 \varepsilon}{\sigma_y(\lambda_0 + 1)} \right).$$

So, for the quadrature rules we can guarantee $|\hat{h}(\mathbf{x}) - h(\mathbf{x})| \leq \varepsilon$ with probability at least $1 - \delta$ as long as

$$D \geq 8M^2(d+1)\sigma_y^2 \left( \frac{\lambda_0 + 1}{\lambda_0^2 \varepsilon} \right)^2 \left[ \frac{2}{1 + \frac{1}{d}} \log \frac{\sigma_y \sigma_p l \kappa \mu (\lambda_0 + 1)}{\lambda_0^2 \varepsilon} + \log \frac{\beta_d}{\delta} \right].$$

## 4 Experiments

We extensively study the proposed method on several established benchmarking datasets: Powerplant, LETTER, USPS, MNIST, CIFAR100 [23], LEUKEMIA [20]. In Section 4.2 we show kernel approximation error across different kernels and number of features. We also report the quality of SVM models with approximate kernels on the same data sets in Section 4.3.

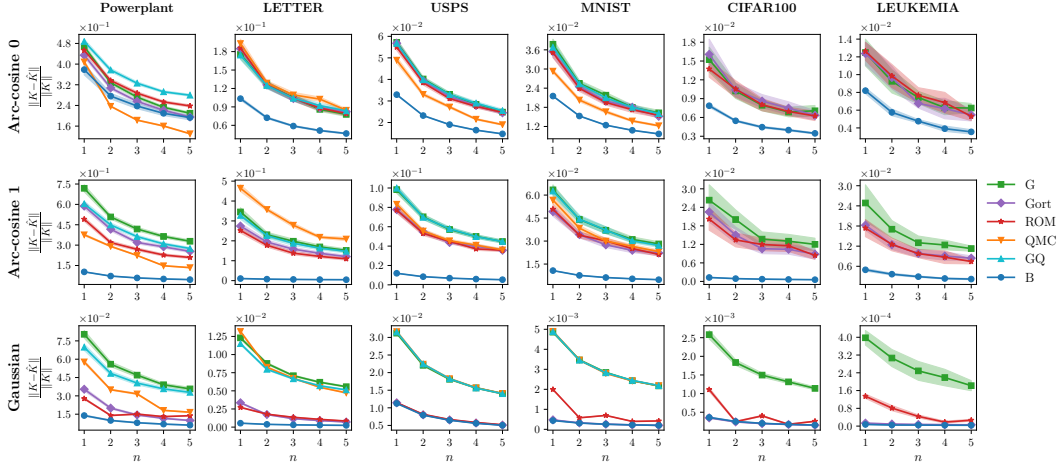

Figure 1: Kernel approximation error across three kernels and 6 datasets. Lower is better. The x-axis represents the factor to which we extend the original feature space, $n = \frac{D}{2(d+1)+1}$, where $d$ is the dimensionality of the original feature space, $D$ is the dimensionality of the new feature space.

## 4.1 Methods

We present a comparison of our method (**B**) with estimators based on a simple Monte Carlo, quasi-Monte Carlo [35] and Gaussian quadratures [11]. The Monte Carlo approach has a variety of ways to generate samples: unstructured Gaussian [29], structured Gaussian [14], random orthogonal matrices (ROM) [10].

**Monte Carlo integration (G, Gort, ROM).** The kernel is estimated as $\hat{k}(\mathbf{x}, \mathbf{y}) = \frac{1}{D}\phi(\mathbf{M}\mathbf{x})\phi(\mathbf{M}\mathbf{y})$, where $\mathbf{M} \in \mathbb{R}^{D \times d}$ is a random weight matrix. For unstructured Gaussian based approximation $\mathbf{M} = \mathbf{G}$, where $\mathbf{G}_{ij} \sim \mathcal{N}(0, 1)$. Structured Gaussian has $\mathbf{M} = \mathbf{G}_{\text{ort}}$, where $\mathbf{G}_{\text{ort}} = \mathbf{D}\mathbf{Q}$, $\mathbf{Q}$ is obtained from RQ decomposition of $\mathbf{G}$, $\mathbf{D}$ is a diagonal matrix with diagonal elements sampled from the $\chi(d)$ distribution. In compliance with the previous work on ROM we use $\mathbf{S}$-Rademacher with three blocks: $\mathbf{M} = \sqrt{d} \prod_{i=1}^{3} \mathbf{S}\mathbf{D}_i$, where $\mathbf{S}$ is a normalized Hadamard matrix and $\mathbb{P}(\mathbf{D}_{ii} = \pm 1) = 1/2$.

**Quasi-Monte Carlo integration (QMC).** Quasi-Monte Carlo integration boasts improved rate of convergence $1/D$ compared to $1/\sqrt{D}$ of Monte Carlo, however, as empirical results illustrate its performance is poorer than that of orthogonal random features [14]. It has larger constant factor hidden under $\mathcal{O}$ notation in computational complexity. For QMC the weight matrix $\mathbf{M}$ is generated as a transformation of quasi-random sequences. We run our experiments with Halton sequences in compliance with the previous work.

**Gaussian quadratures (GQ).** We included subsampled dense grid method from [11] into our comparison as it is the only data-independent approach from the paper that is shown to work well. We reimplemented code for the paper to the best of our knowledge as it is not open sourced.

## 4.2 Kernel approximation

To measure kernel approximation quality we use relative error in Frobenius norm $\frac{\|\mathbf{K} - \hat{\mathbf{K}}\|_F}{\|\mathbf{K}\|_F}$, where $\mathbf{K}$ and $\hat{\mathbf{K}}$ denote exact kernel matrix and its approximation. In line with previous work we run experiments for the kernel approximation on a random subset of a dataset. Table 2 displays the settings for the experiments across the datasets.

Approximation was constructed for different number of $SR$ samples $n = \frac{D}{2(d+1)+1}$, where $d$ is an original feature space dimensionality and $D$ is the new one. For the Gaussian kernel we set hyperparameter $\gamma = \frac{1}{2\sigma^2}$ to the default value of $\frac{1}{d}$ for all the approximants, while the arc-cosine kernels (see definition of arc-cosine kernel in the Supplementary Materials) have no hyperparameters.

We run experiments for each [kernel, dataset, $n$] tuple and plot 95% confidence interval around the mean value line. Figure 1 shows the results for kernel approximation error on LETTER, MNIST, CIFAR100 and LEUKEMIA datasets.

QMC method almost always coincides with RFF except for arc-cosine 0 kernel. It particularly enjoys Powerplant dataset with $d = 4$, i.e. small number of features. Possible explanation for such behaviour can be due to the connection with QMC quadratures. The worst case error for QMC quadratures scales with $n^{-1}(\log n)^d$, where $d$ is the dimensionality and $n$ is the number of sample points [28]. It is worth mentioning that for large $d$ it is also a problem to construct a proper QMC point set. Thus, in higher dimensions QMC may bring little practical advantage over MC. While recent randomized QMC techniques indeed in some cases have no dependence on $d$, our approach is still computationally more efficient thanks to the structured matrices. GQ method as well matches the performance of RFF. We omit both QMC and GQ from experiments on datasets with large $d = [3072, 7129]$ (CIFAR100, LEUKEMIA).

The empirical results in Figure 1 support our hypothesis about the advantages of **SR** quadratures applied to kernel approximation compared to SOTA methods. With an exception of a couple of cases: (Arc-cosine 0, Powerplant) and (Gaussian, USPS), our method displays clear exceeding performance.

## 4.3 Classification/regression with new features

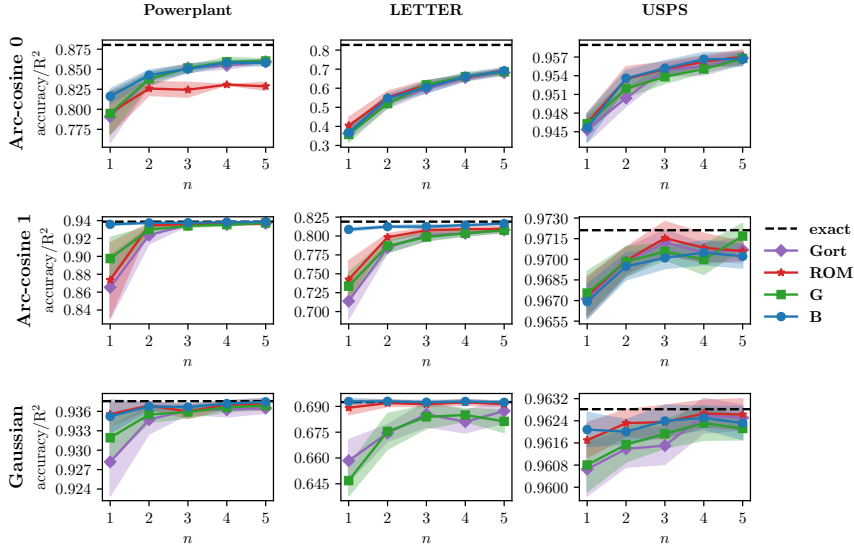

Figure 2: Accuracy/$R^2$ score using embeddings with three kernels on 3 datasets. Higher is better. The x-axis represents the factor to which we extend the original feature space, $n = \frac{D}{2(d+1)+1}$.

We report accuracy and $R^2$ scores for the classification/regression tasks on some of the datasets (Figure 2). We examine the performance with the same setting as in experiments for kernel approximation error, except now we map the whole dataset. We use Support Vector Machines to obtain predictions.

Kernel approximation error does not fully define the final prediction accuracy – the best performing kernel matrix approximant not necessarily yields the best accuracy or $R^2$ score. However, the empirical results illustrate that our method delivers comparable and often superior quality on the downstream tasks.

## 4.4 Walltime experiment

We measure time spent on explicit mapping of features by running each experiment 50 times and averaging the measurements. Indeed, Figure 3 demonstrates that the method scales as theoretically predicted with larger dimensions thanks to the structured nature of the mapping.

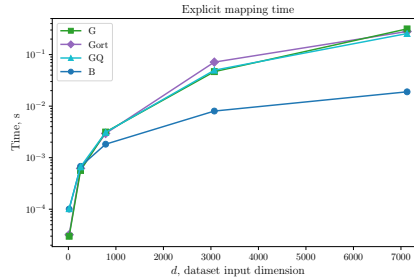

Figure 3: Time spent on explicit mapping. The x-axis represents the 5 datasets with increasing input number of features: LETTER, USPS, MNIST, CIFAR100 and LEUKEMIA.

## 5   Related work

The most popular methods for scaling up kernel methods are based on a low-rank approximation of the kernel using either data-dependent or independent basis functions. The first one includes Nyström method [12], greedy basis selection techniques [31], incomplete Cholesky decomposition [15].

The construction of basis functions in these techniques utilizes the given training set making them more attractive for some problems compared to Random Fourier Features approach. In general, data-dependent approaches perform better than data-independent approaches when there is a gap in the eigen-spectrum of the kernel matrix. The rigorous study of generalization performance of both approaches can be found in [36].

In data-independent techniques, the kernel function is approximated directly. Most of the methods (including the proposed approach) that follow this idea are based on Random Fourier Features [29]. They require so-called weight matrix that can be generated in a number of ways. [24] form the weight matrix as a product of structured matrices. It enables fast computation of matrix-vector products and speeds up generation of random features.

Another work [14] orthogonalizes the features by means of orthogonal weight matrix. This leads to less correlated and more informative features increasing the quality of approximation. They support this result both analytically and empirically. The authors also introduce matrices with some special structure for fast computations. [10] propose a generalization of the ideas from [24] and [14], delivering an analytical estimate for the mean squared error (MSE) of approximation.

All these works use simple Monte Carlo sampling. However, the convergence can be improved by changing Monte Carlo sampling to Quasi-Monte Carlo sampling. Following this idea [35] apply quasi-Monte Carlo to Random Fourier Features. In [37] the authors make attempt to improve quality of the approximation of Random Fourier Features by optimizing sequences conditioning on a given dataset.

Among the recent papers there are works that, similar to our approach, use the numerical integration methods to approximate kernels. While [3] carefully inspects the connection between random features and quadratures, they did not provide any practically useful explicit mappings for kernels. Leveraging the connection [11] propose several methods with Gaussian quadratures. Among them three schemes are data-independent and one is data-dependent. The authors do not compare them with the approaches for random feature generation other than random Fourier features. The data-dependent scheme optimizes the weights for the quadrature points to yield better performance. A closely related work [25] constructs features for kernel approximation by approximating spherical-radial integral and designs QMC points to speed up approximation and reduce memory.

## 6   Conclusion

We propose an approach for the random features methods for kernel approximation, revealing a new interpretation of RFF and ORF. The latter are special cases of the spherical-radial quadrature rules with degrees (1,1) and (1,3) respectively. We take this further and develop a more accurate technique for the random features preserving the time and space complexity of the random orthogonal embeddings.

Our experimental study confirms that for many kernels on the most datasets the proposed approach delivers the best kernel approximation. Additionally, the results showed that the quality of the downstream task (classification/regression) is also superior or comparable to the state-of-the-art baselines.

**Acknowledgments**

This work was supported by the Ministry of Science and Education of Russian Federation as a part of Mega Grant Research Project 14.756.31.0001.

## Footnotes

[1]The code for this paper is available at `https://github.com/maremun/quffka`.

[2]Please see [18] for detailed derivation of the stochastic radial (section 2), spherical (section 3) and spherical radial rules (section 4)

[3]To get $a_0^2 \geq 0$, you need to sample $\rho_j$ two times on average (see Supplementary Materials for details).

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
