[Supplementary Material]

# Quadrature-based features for kernel approximation. Supplementary materials.

**Marina Munkhoeva**[†]     **Yermek Kapushev**[†]     **Evgeny Burnaev**[†]     **Ivan Oseledets**[†,‡]

[†]Skolkovo Institute of Science and Technology
Moscow, Russia

[‡]Institute of Numerical Mathematics of the Russian Academy of Sciences
Moscow, Russia

## 1 Proof of Proposition 3.1

### 1.1 Variance of the degree $(3,3)$ quadrature rule

Let us denote $\mathbf{q} = \begin{pmatrix} \mathbf{x} \\ \mathbf{y} \end{pmatrix} \in \mathcal{X}^2$, $k(\mathbf{q}) = k(\mathbf{x},\mathbf{y})$, $h_j(\mathbf{q}) = d\frac{f_{\mathbf{xy}}(-\rho_j \mathbf{Q}\mathbf{v}_j) + f_{\mathbf{xy}}(\rho_j \mathbf{Q}\mathbf{v}_j)}{2\rho_j^2} - k(\mathbf{q}) = s_j(\mathbf{q}) - k(\mathbf{q})$. Then it is easy to see that $\mathbb{E}h_j(\mathbf{q}) = 0$.

Let us denote $I(\mathbf{q}) = SR^{3,3}_{\mathbf{Q}_1,\rho_1}(f_{\mathbf{xy}})$, $g(\mathbf{q}) = I(\mathbf{q}) - k(\mathbf{x},\mathbf{y})$. Using the above definitions we obtain

$$
\begin{aligned}
\mathbb{V}g(\mathbf{q}) = \mathbb{V}\left(1 - \sum_{j=1}^{d+1}\frac{d}{(d+1)\rho_j^2}\right) + \mathbb{E}\left(\frac{1}{d+1}\sum_{i=1}^{d+1}h_i(\mathbf{q})\right)^2 \\
+ 2cov\left(1 - \sum_{j=1}^{d+1}\frac{d}{(d+1)\rho_j^2}, \frac{1}{d+1}\sum_{i=1}^{d+1}h_i(\mathbf{q})\right).
\end{aligned}
\tag{1}
$$

Variance of the first term

$$
\begin{aligned}
\mathbb{V}\left(1 - \sum_{j=1}^{d+1}\frac{d}{(d+1)\rho_j^2}\right) &= \mathbb{E}\left(1 - \sum_{j=1}^{d+1}\frac{d}{(d+1)\rho_j^2}\right)^2 \\
&= \mathbb{E}\left(1 - \sum_{j=1}^{d+1}\frac{2d}{(d+1)\rho_j^2} + \left(\sum_{j=1}^{d+1}\frac{d}{(d+1)\rho_j^2}\right)^2\right) \\
&= 1 - 2 + \frac{d}{(d+1)(d-2)} + \frac{d}{d+1} = \frac{2}{(d+1)(d-2)}.
\end{aligned}
\tag{2}
$$

Variance of the second term (using independence of $h_i(\mathbf{q})$ and $h_j(\mathbf{q})$ for $i \neq j$)

$$
\mathbb{E}\left(\frac{1}{d+1}\sum_{i=1}^{d+1}h_i(\mathbf{q})\right)^2 = \mathbb{E}\left(\frac{1}{(d+1)^2}\sum_{i,j=1}^{d+1}h_i(\mathbf{q})h_j(\mathbf{q})\right) = \frac{1}{(d+1)^2}\sum_i \mathbb{E}h_i(\mathbf{q})^2 = \frac{\mathbf{E}h_1(\mathbf{q})^2}{d+1}.
\tag{3}
$$

Variance of the last term (using Cauchy-Schwarz inequality)

$$cov\left(1 - \sum_{j=1}^{d+1} \frac{d}{(d+1)\rho_j^2}, \frac{1}{d+1}\sum_{i=1}^{d+1} h_i(\mathbf{q})\right) = \mathbb{E}\left[\left(1 - \sum_{j=1}^{d+1} \frac{d}{(d+1)\rho_j^2}\frac{1}{d+1}\right)\sum_{i=1}^{d+1} h_i(\mathbf{q})\right]$$

$$= -\mathbb{E}\frac{d}{d+1}\sum_{i,j=1}^{d+1}\frac{h_i(\mathbf{q})}{\rho_j^2}$$

$$\leq \frac{1}{d+1}\sum_{i=1}^{d+1}\sqrt{\mathbb{E}\frac{1}{\rho_i^4}}\sqrt{\mathbb{E}h_i(\mathbf{q})^2}$$

$$= \sqrt{\frac{\mathbb{E}h_1(\mathbf{q})^2}{d(d-2)}}. \tag{4}$$

Now, let us upper bound term $\mathbb{E}h_1(\mathbf{q})^2$

$$\mathbb{E}h_1(\mathbf{q})^2 = \mathbb{E}\left(\frac{d\phi(\mathbf{w}^\top\mathbf{x})\phi(\mathbf{w}^\top\mathbf{y})}{\rho^2}\right)^2 - k(\mathbf{q})^2 \leq \frac{d\kappa^4}{d-2}.$$

Using this expression and plugging (2), (3), (4) into (1) we obtain

$$\mathbb{V}\left[\frac{1}{n}\sum_{i=1}^{n} SR_{\mathbf{Q}_i,\rho_i}^{3,3}(f_{\mathbf{xy}})\right] \leq \frac{2}{n(d+1)(d-2)} + \frac{d\kappa^4}{n(d+1)(d-2)} + \frac{1}{n}\sqrt{\frac{d\kappa^4}{d(d-2)^2}} \leq$$

$$\leq \frac{2}{n(d+1)(d-2)} + \frac{d\kappa^4}{n(d+1)(d-2)} + \frac{\kappa^2}{n(d-2)} \leq \frac{2+\kappa^4+\kappa^2}{n(d-2)}. \tag{5}$$

and it concludes the proof.

## 1.2  Error probability

The proof strategy closely follows that of [3]; we just use Chebyshev-Cantelli ineqaulity instead of Hoeffding's and Bernstein inequalities and all the expectations are calculated according to our quadrature rules.

Let $\mathbf{q} = \begin{pmatrix}\mathbf{x}\\\mathbf{y}\end{pmatrix} \in \mathcal{X}^2$, $\mathcal{X}^2$ is compact set in $\mathbb{R}^{2d}$ with diameter $\sqrt{2}l$, so we can cover it with an $\varepsilon$-net using at most $T = (2\sqrt{2}l/r)^{2d}$ balls of radius $r$. Let $\{\mathbf{q}_i\}_{i=1}^{T}$ denote their centers, and $L_g$ be the Lipschitz constant of $g(\mathbf{q}) : \mathbb{R}^{2d} \to \mathbb{R}$. If $|g(\mathbf{q}_i)| < \varepsilon/2$ for all $i$ and $L_g < \varepsilon/(2r)$, then $g(\mathbf{q}) < \varepsilon$ for all $\mathbf{q} \in \mathcal{X}^2$.

### 1.2.1  Regularity Condition

Similarly to [3] (regularity condition section in appendix) it can be proven that $\mathbb{E}\nabla g(\mathbf{q}) = \nabla\mathbb{E}g(\mathbf{q})$.

### 1.2.2  Lipschitz Constant

Since $g$ is differentiable, $L_g = \|\nabla g(\mathbf{q}^*)\|$, where $\mathbf{q}^* = \arg\max_{\mathbf{q}\in\mathcal{X}^2}\|\nabla g(\mathbf{q})\|$. Via Jensen's inequality $\mathbb{E}\|\nabla h(\mathbf{q})\| \geq \|\mathbb{E}\nabla h(\mathbf{q})\|$. Then using independence of $h_i(\mathbf{q})$ and $h_j(\mathbf{q})$ for $i \neq j$

$$\mathbb{E}[L_g]^2 = \mathbb{E}\left[\|\nabla I(\mathbf{q}^*) - k(\mathbf{q}^*)\|^2\right] = \mathbb{E}\left[\left\|\frac{1}{d+1}\sum_{i=1}^{d+1}\nabla h_i(\mathbf{q}^*)\right\|^2\right] = \mathbb{E}\left[\frac{1}{d+1}\|\nabla h_1(\mathbf{q}^*)\|^2\right] =$$

$$= \frac{1}{d+1}\mathbb{E}_{\mathbf{q}^*}\left[\mathbb{E}\|\nabla s_1(\mathbf{q}^*)\|^2 - 2\|\nabla k(\mathbf{q}^*)\|\mathbb{E}\|\nabla s_1(\mathbf{q}^*)\| + \|\nabla k(\mathbf{q}^*)\|^2\right] \leq$$

$$\leq \frac{1}{d+1}\mathbb{E}\left[\|\nabla s_1(\mathbf{q}^*)\|^2 - \|\nabla k(\mathbf{q}^*)\|^2\right] \leq \frac{1}{d+1}\mathbb{E}\|\nabla s_1(\mathbf{q}^*)\|^2 =$$

$$= \frac{1}{d+1}\mathbb{E}\left[\|\nabla_{\mathbf{x}^*}s_1(\mathbf{q}^*)\|^2 + \|\nabla_{\mathbf{y}^*}s_1(\mathbf{q}^*)\|^2\right] \leq \frac{2d^2\kappa^2\mu^2\sigma_p^2}{d+1}\mathbb{E}\frac{1}{\rho_1^2} = \frac{2d\kappa^2\mu^2\sigma_p^2}{d+1},$$

where $|\phi'(\cdot)| \leq \mu$. Then using Markov's inequality we obtain

$$\mathbb{P}(L_g \geq \frac{\varepsilon}{2r}) \leq 8\frac{d}{d+1}\left(\frac{\sigma_p r \kappa \mu}{\varepsilon}\right)^2$$

### 1.2.3 Anchor points

Let us upper bound the following probability

$$\mathbb{P}\left(\bigcup_{i=1}^{T}|g(\mathbf{q}_i)| \geq \frac{1}{2}\varepsilon\right) \leq T\mathbb{P}\left(|g(\mathbf{q}_i)| \geq \frac{1}{2}\varepsilon\right).$$

Let us rewrite the function $g(\mathbf{q})$

$$g(\mathbf{q}) = 1 - \frac{1}{d+1}\sum_{i=1}^{d+1}\frac{d}{\rho_i^2} + \frac{1}{d+1}\sum_{i=1}^{d+1}\frac{d\phi_{\mathbf{q}}(\rho_i \mathbf{z}_i)}{\rho_i^2} - k(\mathbf{q}) = \frac{1}{d+1}\sum_{i=1}^{d+1}\left(\frac{d(1-\phi_{\mathbf{q}}(\rho_i \mathbf{z}_i))}{\rho_i^2} + 1 - k(\mathbf{q})\right),$$

where $\phi_{\mathbf{q}}(\rho_i \mathbf{z}_i) = \frac{f_{\mathbf{xy}}(-\rho_j \mathbf{Q}\mathbf{v}_j) + f_{\mathbf{xy}}(\rho_j \mathbf{Q}\mathbf{v}_j)}{2\rho_j^2}$. Let us suppose that $\left|\frac{1-\phi_{\mathbf{q}}(\rho \mathbf{z})}{\rho^2}\right| \leq M$. Then we can apply Hoeffding's inequality

$$\mathbb{P}(|g(\mathbf{q})| \geq \frac{1}{2}\varepsilon) \leq 2\exp\left(-\frac{2D\frac{1}{4}\varepsilon^2}{(M-(-M))^2}\right) = 2\exp\left(-\frac{D\varepsilon^2}{8M^2}\right)$$

### 1.2.4 Optimizing over $r$

Now the probability of $\sup_{\mathbf{q}\in\mathcal{X}^2}|g(\mathbf{q})| \leq \varepsilon$ takes the form

$$p = \mathbb{P}\left(\sup_{\mathbf{q}\in\mathcal{X}^2}|g(\mathbf{q})| \leq \varepsilon\right) \geq 1 - \kappa_1 r^{-2d} - \kappa_2 r^2,$$

where $\kappa_1 = 2\left(2\sqrt{2}l\right)^{2d}\exp\left(-\frac{D\varepsilon^2}{8M^2}\right)$, $\kappa_2 = \frac{8d}{d+1}\left(\frac{\kappa\mu\sigma_p}{\varepsilon}\right)^2$. Maximizing this probability over $r$ gives us the following bound

$$\mathbb{P}\left(\sup_{\mathbf{q}\in\mathcal{X}^2}|g(\mathbf{q})| \geq \varepsilon\right) \leq \left(d^{\frac{-d}{d+1}} + d^{\frac{1}{d+1}}\right)2^{\frac{6d+1}{d+1}}\left(\frac{d}{d+1}\right)^{\frac{d}{d+1}}\left(\frac{\sigma_p l \kappa \mu}{\varepsilon}\right)^{\frac{2d}{d+1}}\exp\left(-\frac{D\varepsilon^2}{8M^2(d+1)}\right).$$

For RBF kernel $\kappa = \mu = 1$, $M = \frac{1}{2}$, so we obtain the following bound

$$\mathbb{P}\left(\sup_{\mathbf{q}\in\mathcal{X}^2}|g(\mathbf{q})| \geq \varepsilon\right) \leq \left(d^{\frac{-d}{d+1}} + d^{\frac{1}{d+1}}\right)2^{\frac{6d+1}{d+1}}\left(\frac{d}{d+1}\right)^{\frac{d}{d+1}}\left(\frac{\sigma_p l}{\varepsilon}\right)^{\frac{2d}{d+1}}\exp\left(-\frac{D\varepsilon^2}{2(d+1)}\right).$$

Let us compare it with the bound for RFF

$$\mathbb{P}\left(\sup_{\mathbf{q}\in\mathcal{X}^2}|g(\mathbf{q})| \geq \varepsilon\right) \leq \left(d^{\frac{-d}{d+1}} + d^{\frac{1}{d+1}}\right)2^{\frac{5d+1}{d+1}}3^{\frac{d}{d+1}}\left(\frac{\sigma_p l}{\varepsilon}\right)^{\frac{2d}{d+1}}\exp\left(-\frac{D\varepsilon^2}{32(d+1)\alpha_{\varepsilon}'}\right).$$

## 2 Butterfly matrices

For orthogonal matrix $\mathbf{Q}$ in quadrature rules the so called butterfly matrix is used. As it happens to be a product of butterfly structured factors, a matrix of this type conveniently possesses the property of fast multiplication. For $d = 4$ an example of butterfly orthogonal matrix is

$$\mathbf{B}^{(4)} = \begin{bmatrix} c_1 & -s_1 & 0 & 0 \\ s_1 & c_1 & 0 & 0 \\ 0 & 0 & c_3 & -s_3 \\ 0 & 0 & s_3 & c_3 \end{bmatrix}\begin{bmatrix} c_2 & 0 & -s_2 & 0 \\ 0 & c_2 & 0 & -s_2 \\ s_2 & 0 & c_2 & 0 \\ 0 & s_2 & 0 & c_2 \end{bmatrix} = \begin{bmatrix} c_1 c_2 & -s_1 c_2 & -c_1 s_2 & s_1 s_2 \\ s_1 c_2 & c_1 c_2 & -s_1 s_2 & -c_1 s_2 \\ c_3 s_2 & -s_3 s_2 & c_3 c_2 & -s_3 c_2 \\ s_3 s_2 & c_3 s_2 & s_3 c_2 & c_3 c_2 \end{bmatrix}.$$

Figure 1: (a) Butterfly orthogonal matrix factors for $d = 16$. (b) Sparsity pattern for $\mathbf{BPBPBP}$ (left) and $\mathbf{B}$ (right), $d = 15$.

**Definition 2.1.** Let $c_i = \cos\theta_i$, $s_i = \sin\theta_i$ for $i = 1, \ldots, d-1$ be given. Assume $d = 2^k$ with $k > 0$. Then an orthogonal matrix $\mathbf{B}^{(d)} \in \mathbb{R}^{d \times d}$ is defined recursively as follows

$$\mathbf{B}^{(2d)} = \begin{bmatrix} \mathbf{B}^{(d)} c_d & -\mathbf{B}^{(d)} s_d \\ \hat{\mathbf{B}}^{(d)} s_d & \hat{\mathbf{B}}^{(d)} c_d \end{bmatrix}, \quad \mathbf{B}^{(1)} = 1,$$

where $\hat{\mathbf{B}}^{(d)}$ is the same as $\mathbf{B}^{(d)}$ with indexes $i$ shifted by $d$, e.g.

$$\mathbf{B}^{(2)} = \begin{bmatrix} c_1 & -s_1 \\ s_1 & c_1 \end{bmatrix}, \quad \hat{\mathbf{B}}^{(2)} = \begin{bmatrix} c_3 & -s_3 \\ s_3 & c_3 \end{bmatrix}.$$

Matrix $\mathbf{B}^{(d)}$ by vector product has computational complexity $O(d \log d)$ since $\mathbf{B}^{(d)}$ has $\lceil \log d \rceil$ factors and each factor requires $O(d)$ operations. Another advantage is space complexity: $\mathbf{B}^{(d)}$ is fully determined by $d - 1$ angles $\theta_i$, yielding $O(d)$ memory complexity.

The randomization is based on the sampling of angles $\theta$. We follow [2] algorithm that first computes a uniform random point $\mathbf{u}$ from $U_d$. It then calculates the angles by taking the ratios of the appropriate $\mathbf{u}$ coordinates $\theta_i = \frac{u_i}{u_{i+1}}$, followed by computing cosines and sines of the $\theta$'s. One can easily define butterfly matrix $\mathbf{B}^{(d)}$ for the cases when $d$ is not a power of two.

## 2.1 Not a power of two

We discuss here the procedure to generate butterfly matrices of size $d \times d$ when $d$ is not a power of 2.

Let the number of butterfly factors $k = \lceil \log d \rceil$. Then $\mathbf{B}^{(d)}$ is constructed as a product of $k$ factor matrices of size $d \times d$ obtained from $k$ matrices used for generating $\mathbf{B}^{(2^k)}$. For each matrix in the product for $\mathbf{B}^{(2^k)}$, we delete the last $2^k - d$ rows and columns. We then replace with 1 every $c_i$ in the remaining $d \times d$ matrix that is in the same column as deleted $s_i$.

For the cases when $d$ is not a power of two, the resulting $\mathbf{B}$ has deficient columns with zeros (Figure 1b, right), which introduces a bias to the integral estimate. To correct for this bias one may apply additional randomization by using a product $\mathbf{BP}$, where $\mathbf{P} \in \{0, 1\}^{d \times d}$ is a permutation matrix. Even better, use a product of several $\mathbf{BP}$'s: $\widetilde{\mathbf{B}} = (\mathbf{BP})_1 (\mathbf{BP})_2 \ldots (\mathbf{BP})_t$. We set $t = 3$ in the experiments.

## 3 Remarks on quadrature rules

**Even functions.** We note here that for specific functions $f_{\mathbf{xy}}(\mathbf{w})$ we can derive better versions of $SR$ rule by taking on advantage of the knowledge about the integrand. For example, the Gaussian kernel has $f_{\mathbf{xy}}(\mathbf{w}) = \cos(\mathbf{w}^\top(\mathbf{x} - \mathbf{y}))$. Note that $f$ is even, so we can discard an excessive term in the summation in degree $(3, 3)$ rule, since $f(\mathbf{w}) = f(-\mathbf{w})$, i.e $SR^{3,3}$ rule reduces to

$$SR^{3,3}_{\mathbf{Q},\rho}(f) = \left(1 - \sum_{j=1}^{d+1} \frac{d}{(d+1)\rho_j^2}\right) f(\mathbf{0}) + \frac{d}{d+1} \sum_{j=1}^{d+1} \frac{f(\rho_j \mathbf{Q} \mathbf{v}_j)}{\rho_j^2}. \tag{6}$$

**Obtaining a proper $\rho$.** It may be the case when sampling $\rho$ that $1 - \sum_{j=1}^{d+1} \frac{d}{(d+1)\rho_j^2} < 0$ which results in complex $a_0$ term. Simple solution is just to resample $\rho_j$ to satisfy the non-negativity of the expression. According to central limit theorem $\sum_{j=1}^{d+1} \frac{d}{(d+1)\rho_j^2}$ tends to normal random variable with mean 1 and variance $\frac{1}{d+1}\frac{2}{d-2}$. The probability that this values is non-negative equals $p = \mathbb{P}(1 - \sum_{j=1} \frac{d}{(d+1)\rho^2} \geq 0) \rightsquigarrow \frac{1}{2}$. The expectation of number of resamples needed to satisfy non-negativity constraint is $\frac{1}{p}$ tends to 2.

## 4 Arc-cosine kernels

Arc-cosine kernels were originally introduced by [1] upon studying the connections between deep learning and kernel methods. The integral representation of the $b^{th}$-order arc-cosine kernel is

$$k_b(\mathbf{x}, \mathbf{y}) = 2 \int_{\mathbb{R}^n} \Theta(\mathbf{w}^\top \mathbf{x}) \Theta(\mathbf{w}^\top \mathbf{y}) (\mathbf{w}^\top \mathbf{x})^b (\mathbf{w}^\top \mathbf{y})^b p(\mathbf{w}) d\mathbf{w},$$

$$k_b(\mathbf{x}, \mathbf{y}) = 2 \int_{\mathbb{R}^d} \phi_b(\mathbf{w}^\top \mathbf{x}) \phi_b(\mathbf{w}^\top \mathbf{y}) p(\mathbf{w}) d\mathbf{w},$$

where $\phi_b(\mathbf{w}^\top \mathbf{x}) = \Theta(\mathbf{w}^\top \mathbf{x})(\mathbf{w}^\top \mathbf{x})^b$, $\Theta(\cdot)$ is the Heaviside function and $p$ is the density of the standard Gaussian distribution. Such kernels can be seen as an inner product between the representation produced by infinitely wide single layer neural network with random Gaussian weights. They have closed form expressions in terms of the angle $\theta = \cos^{-1}\left(\frac{\mathbf{x}^\top \mathbf{y}}{\|\mathbf{x}\|\|\mathbf{y}\|}\right)$ between $\mathbf{x}$ and $\mathbf{y}$.

Arc-cosine kernel of $0^{th}$-order shares the property of mapping the input on the unit hypersphere with RBF kernels, while order 1 arc-cosine kernel preserves the norm as linear kernel (Gram matrix on original features):

These expressions for $0^{th}$-order and $1^{st}$-order arc-cosine kernels are given by

$$k_0(\mathbf{x}, \mathbf{y}) = 1 - \frac{\theta}{\pi}, \qquad k_1(\mathbf{x}, \mathbf{y}) = \frac{\|\mathbf{x}\|\|\mathbf{y}\|}{\pi}(\sin\theta + (\pi - \theta)\cos\theta).$$

The 0-order arc-cosine kernel is given by $k_0(\mathbf{x}, \mathbf{y}) = 1 - \frac{\theta}{\pi}$, the 1-order kernel is given by $k_1(\mathbf{x}, \mathbf{y}) = \frac{\|\mathbf{x}\|\|\mathbf{y}\|}{\pi}(\sin\theta + (\pi - \theta)\cos\theta)$.

Let $\phi_0(\mathbf{w}^\top \mathbf{x}) = \Theta(\mathbf{w}^\top \mathbf{x})$, $\phi_1(\mathbf{w}^\top \mathbf{x}) = \max(0, \mathbf{w}^\top \mathbf{x})$. We now can rewrite the integral representation as follows:

$$k_b(\mathbf{x}, \mathbf{y}) = 2 \int_{\mathbb{R}^d} \phi_b(\mathbf{w}^\top \mathbf{x}) \phi_b(\mathbf{w}^\top \mathbf{y}) p(\mathbf{w}) d\mathbf{w} \approx \frac{2}{n} \sum_{i=1}^{n} SR^{3,3}_{\mathbf{Q}_i, \boldsymbol{\rho}_i}.$$

For arc-cosine kernel of order 0 the value of the function $\phi_0(0) = \Theta(0) = 0.5$ results in

$$SR^{3,3}_{\mathbf{Q}, \rho}(f) = 0.25\left(1 - \sum_{j=1}^{d+1} \frac{d}{(d+1)\rho_j^2}\right) + \frac{d}{d+1}\sum_{j=1}^{d+1} \frac{f(\rho_j \mathbf{Q} \mathbf{v}_j) + f(-\rho_j \mathbf{Q} \mathbf{v}_j)}{2\rho^2}.$$

In the case of arc-cosine kernel of order 1, the value of $\phi_1(0)$ is 0 and the $SR^{3,3}$ rule reduces to

$$SR^{3,3}_{\mathbf{Q}, \rho}(f) = \frac{d}{d+1}\sum_{j=1}^{d+1} \frac{f(|\rho \mathbf{Q} \mathbf{v}_j|)}{2\rho_j^2}.$$