[Reviews · NeurIPS 2018]

Reviewer 1



*** UPDATE *** Thank you for your response. Unfortunately I could not easily see where, in the Owen 2017 reference that you suggested, it was shown that the extensible constructions do not need to worry about sub-sequences. Indeed, I am still under the impression that at iteration n of a QMC method, we add on a term of O(1/n), so that the integration error cannot be o(1/n) at all n? This seems to be the position of "A constraint on extensible quadrature rules" (Owen, 2016); see the first sentence of section 2. Hopefully this point can be properly resolved in a revision to the main text. Apart from this, it seems that the paper is good enough for NIPS. *** This paper is one of several recent attempts to reduce the integration error in the original random Fourier features work. Although I have some reservations about the limited experimental comparison, the authors should be commended for putting well-documented code on GitHub. The main points of concern that I had are as follows: - I do not understand why the authors restrict to stochastic integration procedures, when deterministic cubature methods (such as Gauss-Hermite quadrature) are simpler and already widely used. Some motivation/justification would be useful. - on p2, in the definition of \psi(x), I think a_1,...,a_D should be used instead of a,...,a. - In the definition of stochastic spherical-radial rules, there is a sum over j but the summand does not depend on j. This is probably an error. - It is unclear where the statement of Proposition 3.2 actually ends; I think the authors have not used the standard latex amsthm environment (so the statement is not in italic font). - I am concerned about the use of the Halton sequence: It is known that *only* the sequences of length 2^n have good approximation properties for numerical cubature, and it is therefore unfair to implement QMC based on a number of points that is not a power of 2. It looks, though I am not certain, that the authors have not acknowledged that. - I didn't understand the comment on p7 explaining why QMC and GQ methods cannot be used in high-dimensional problems. At least, if you talk to a QMC researcher, they will say that the main purpose of QMC is to address high-dimensional problems. Perhaps some politically judicious rephrasing would be useful. - As mentioned above, I was disappointed to see that there was not an empirical comparison with [11]. This was probably the main weakness of the paper.

Reviewer 2



The paper proposes a novel kernel approximation method by utilizing quadrature rules. Similar to random Fourier features, the proposed approach targets shift invariant kernels and pointwise non-linear Gaussian kernels. The spherical radial quadrature rules are applied rather than simple monte carlo (as is the case in RFF) and are shown to improve the constant terms in the asymptotic convergence to exact inference as the number of samples tends to infinity. Further, conditions for probabilistic epsilon-delta bounds are derived for the absolute error on the results of ridge regression in the kernel space. RFFs and Orthogonal Random Features are both shown to special cases of the general approach. The authors take advantage of Butterfly matrices which speedup the approximation. Finally the authors perform experimental results on a range of datasets and show speed and accuracy increases. At a high level I thought the paper was quite dense with content and the supplementary material was necessary to go through in detail in order to follow a number of the claims in the paper. In the supplementary material, however, the authors took advantage of the extra space and made it a lot easier to follow and understand more clearly. In comparison to, for example, RFFs the method is a little more complex by nature as they are not using the trivial simple monte carlo in Fourier space approach. However, this appears come come with great advantage from a performance point of view. Fortunately it appears the authors have made available the code via a GitHub repo (which should probably be left as an anonymous link until after the review in future in order to preserve fully double blind reviews). In lines 112-114, the authors also explicitly describe how to use the approach for the commonly used Gaussian kernel and others. The results show the normed relative error of an approximate Gaussian, Arccos 0 and Arccos 1 kernel over a range of standard datasets. The proposed method consistently outperforms competing approaches. However, I would suggest clarifying that ‘B’ represents the proposed approach. Not only is is there reduced error in the kernel matrix but they show empirically that this has a knock on effect to the accuracy. Again I would note that the authors may want to be consistent between the figures in terms of naming convention (Arccos vs Arc-cosine) but that is only a minor point. The only part of the paper which did not come across as thorough, be it a minor part, was the timed experiments. One would expect the authors to run the timing multiple times so that the effects of randomness due to processor scheduling etc are not causing any effect. That said for d=3000:7000 there appears to be an order of magnitude difference between explicit mapping times. Overall I think the paper was well written, presents an interesting addition to the literature and opens the door for future research which takes advantage of other quadrature rules and classes of kernels.

Reviewer 3



Post rebuttal: Thanks for the feedback. Summary of the paper: This paper proposes a novel class of random feature maps that subsumes the standard random Fourier features by Rahimi and orthogonal random features by [16]. The approach is based on spherical radial quadrature rules by Genz and Monahan [18]. Some theoretical guarantees are provided, and impressive experimental results are reported. ~~~ Quality ~~~ Strength: Extensive experimental results have been conducted, showing impressive empirical performance of the proposed approach. Also some theoretical guarantees are provided. Weakness: Theoretical error-bounds seem not very tight, and it is theoretically not very clear why the proposed works well; see the detailed comments below. I guess this would be because of the property of spherical radial rules being exact for polynomials up to a certain order. ~~~ Clarity ~~~ Strength: This paper is basically written well, and connections to the existing approaches are discussed concisely. Weakness: It would be possible to explain theoretical properties of spherical quadrature rules more explicitly, and this would help understand the behaviour of the proposed method. ~~~ Originality ~~~ Strength: I believe the use of spherical radial quadrature is really novel. Weakness: It seems that theoretical arguments basically follow those of [32]. ~~~ Significance ~~~ Strength: The proposed framework subsumes the standard random Fourier features by Rahimi and orthogonal random features by [16]; this is very nice, as it provides a unifying framework for existing approaches to random feature maps. Also the experimental results are impressive (in particular for approximating kernel matrices). The proposed framework would indicate a new research direction, encouraging the use of classical quadrature rules in the numerical analysis literature. %%% Detailed comments %%% ++ On the properties of spherical radial rules ++ You mentioned that the weights of a spherical radial rule [18] are derived so that they are exact for polynomials up to a certain order and unbiased for other functions. We have the following comments: - Given that [18] is a rather minor work (guessing from the number of citations; I was not aware of this work), it would be beneficial for the reader to state this property in the form of Theorem or something similar, at least in Appendix. From the current presentation it is not very clear what exactly the this property means mathematically. - Are spherical radial rules related to Gauss-Hermite quadrature, which are (deterministic) quadrature rules that are exact for polynomials up to a certain order and deals with Gaussian measures? ++ Line 109 and Footnote 3 ++ From what you mentioned that with some probability $a_0$ can be an imaginary number. What should one do if this is the case? ++ Proposition 3.1 ++ The upper bound is polynomial in $\epsilon$, which seems quite loose, compared to the exponential bound derived in the following paper: Optimal rates for random Fourier features B. K. Sriperumbudur and Z. Szabo Neural Information Processing Systems, 2015 ++ Proposition 3.2 ++ What is $\lambda$? Where is this defined?